REGISTERED REPORT PROTOCOL

# Electromyographic parameters for treatment of pelvic floor disorders in pregnant and postpartum women: A review protocol

**Alethéa Cury Rabelo Leitão**[1]*, **Silvia Oliveira Ribeiro Lira**[2,3], **Elizabel de Souza Ramalho Viana**[1]

1 Department of Physical Therapy, Universidade Federal do Rio Grande do Norte, Natal, Rio Grande do Norte, Brazil, 2 Federal University of Rio Grande do Norte, Natal, Rio Grande do Norte, Brazil, 3 Faculdade de Ciências da Saúde do Trairi (FACISA/UFRN), Santa Cruz, Rio Grande do Norte, Brazil

* ale_cury@yahoo.com.br

This is a Registered Report and may have an associated publication; please check the article page on the journal site for any related articles.

## Abstract

Electromyography is a widely used instrument in clinical practice to evaluate and treat pelvic floor disorders in pregnant and postpartum women. The objective of this study is to analyze the scientific evidence on the electromyography parameters used for treatment of pelvic floor disorders in pregnant women in any gestational week and postpartum women up to 12 months after delivery. A systematic review of randomized controlled experimental studies (clinical trials) and quasi-experimental studies in English, Portuguese or Spanish, which used electromyography as an intervention for treatment of pelvic floor disorders in pregnant or postpartum women up to 12 months after delivery will be performed in online databases (Scopus, Medline, Pedro, Scielo and Pubmed),. Risk of bias assessment will be performed using Cochrane group tools. The Rob 2.0 tool will be used for experimental studies and the Robins-I tool for non-experimental studies. The protocol was registered in PROSPERO (no.433510). The quality of the evidence will be analyzed using the GRADE System Methodological Guide and the systematic review structure will be performed according to the Preferred Reporting Items for Systematic Reviews and Meta-Analyses (PRISMA) guidelines.

## Introduction

### Description of the condition

It is consensus that the risk factors for developing pelvic floor disorders in women are related to the pregnancy and delivery period [1]. The literature shows that 50% of women lose some support functionality of the pelvic floor muscles (PFM) due to childbirth, and these injuries increase by an average of 20% in women who had vaginal delivery [2].

The pelvic floor of these women will be overloaded during pregnancy due to enlargement of the gravid uterus from anatomical and physiological changes [3]. Deficiency in the pelvic floor muscles (PFM) can lead to developing stress urinary incontinence (SUI), fecal

**Data Availability Statement:** All relevant data are within the paper and its Supporting Information files.

**Funding:** The authors received no specific funding for this work.

**Competing interests:** The authors have declared that no competing interests exist.

incontinence (FI), and pelvic organ prolapse (POP) [3, 4]. These pelvic floor dysfunctions (PFD) negatively impact the quality of life in women, creating a significant social problem [4].

A recent study indicates that women who had vaginal deliveries exhibited greater pelvic floor muscle weakness compared to those who had cesarean sections. The proposed treatment utilized various electromyographic parameters to rehabilitate pelvic floor muscle dysfunctions caused by muscle fiber rupture during childbirth [5].

Pelvic floor muscle training (PFMT) is the first-line treatment for dysfunctions of this musculature [6]. This treatment is based on increasing strength, endurance, maintaining muscle contraction for a long period of time, muscle coordination, adherence to and motivation for the training program [7, 8].

The literature includes proposed protocols that supplement pelvic floor muscle training (PFMT) with surface electromyography (sEMG) for strengthening the pelvic floor muscles in treatment of stress urinary incontinence (SUI) in pregnant women [9, 10].

It is necessary to emphasize the importance of defining the electromyographic parameters used in treatment of pelvic floor muscle dysfunctions (PFMD). Standardizing these parameters will enable professionals to reproduce and validate results in scientific studies, thereby contributing to develop evidence-based therapeutic plans. Electromyography can be an adjunct to pelvic floor muscle training for treatment of urinary disorders in pregnant and postpartum women.

## Description of the intervention

Surface electromyography (sEMG) is a therapeutic modality applied to treat pelvic floor muscle (PFM) dysfunctions [1, 11]. sEMG is used to capture and read the myoelectric activity of the PFM during the treatment of pelvic floor dysfunctions [12], providing visual or auditory biofeedback (EMG-BF).

A 2021 meta-analysis with more than four thousand women found that the use of electromyography combined with conservative treatment of the pelvic floor muscles has better results than the isolated treatment [1]. In a meta-analysis of 11 studies, Nunes *et al.* (2019) [13] concluded that PFMT combined with sEMG offers therapeutic benefits over other interventions in treatment of female SUI [9]. Additionally, a systematic review by López-Liria *et al.* (2019) [14] analyzed the efficacy of different techniques in treatment of female SUI. The findings showed that sEMG, when combined with PFMT, is more effective in treatment of pelvic floor muscle dysfunctions (PFD) compared to other techniques analyzed. This effectiveness is attributed to the observed increase in muscle contraction.

There are several methods for evaluating and treatment of the functionality of the pelvic floor muscles, such as: digital palpation, manometry, ultrasound, electromyography and magnetic resonance imaging [6, 15].

sEMG provides important data on baseline tone, type of contraction, signal amplitude, changes in signal spectrum, amplitude variability, and recovery time of the pelvic floor muscles (PFM) [11]. These data are essential for describing muscle functionality during maximal or isometric contraction exercises and muscle behavior during rest [16].

## How the intervention will work

sEMG is an easily applicable, versatile, and comprehensible tool [3]. Clinical studies have demonstrated that conservative training with sEMG enables professionals and patients to accurately and objectively observe the contraction and relaxation of pelvic floor muscles (PFM) [11]. This facilitates neuromuscular learning and allows for more effective rehabilitation [8].

Electromyography can be used alone or in combination with conservative treatment. It enables indicating the activity of the pelvic floor muscles in relation to rest during contraction and during relaxation [13, 17].

Clinical studies have evaluated the bioelectric activity of this muscle group using electromyography [18]. This resource has been used as an adjunct to conservative training and enables the physiotherapist to correctly and objectively observe the contraction and relaxation of the pelvic floor muscles. Therefore, it facilitates neuromuscular learning and more assertively performs rehabilitation [13, 17].

More recent reviews which analyzed the effectiveness of conservative treatment with and without electromyography in relation to pelvic floor strength, urinary incontinence score and quality of sexual life excluded pregnant and postpartum women from the intervention groups to analyze these effects, leaving a gap in the literature for this population [14, 19, 20].

Knowledge about data on frequency, intensity and type of muscle contraction can more assertively determine goals and therapeutic conduct. The electromyographic parameters of the pelvic floor muscles guide the health professional in choosing the best strategies for the successful treatment of PFM disorders in pregnant and postpartum women. However, due to the scarcity of studies in this population, a systematic investigation of the literature on the conducts carried out and their effects so far is necessary.

## Why is this review important?

It is important to have a better understanding of the electromyography parameters most used in treatment of pelvic floor muscle disorders in pregnant and postpartum women. Identifying electromyography data can be a reference for elaborating pelvic floor rehabilitation procedures, such as data used for muscle strengthening or endurance training that improve vaginal occlusion in cases of stress urinary incontinence (SUI), and thereby contribute to prevent and treat dysfunctions in pelvic floor muscles. Furthermore, previous systematic reviews have not analyzed the most appropriate therapy for this population when using EMG alone or in combination in this population.

Therefore, this study aims to analyze the scientific evidence about electromyographic parameters in treatment of pelvic floor dysfunctions in pregnant and postpartum women.

## Materials and methods

### Protocol and guidelines

This review will be conducted following the Preferred Reporting Items for Systematic Reviews and Meta-Analyses (PRISMA) guidelines [21]. The protocol was registered with PROSPERO (no.433510) and will be conducted between August 2023 and October 2024.

### Type of studies

The review will include randomized controlled experimental studies (RCTs) and non-experimental studies (nRCTs), in English, Portuguese or Spanish, using sEMG in pregnant or postpartum women for treatment of pelvic floor dysfunctions (IUE) which analyzed baseline tone, contraction and relaxation capacity.

### Type of participants

Studies will include pregnant women (at any gestational week) and postpartum women (up to 12 months postpartum) with pelvic floor dysfunction (PFD) who were treated with EMG.

## Type of interventions

In addition, studies which used electromyography as an instrument, in isolation or in combination, for evaluating or treatment of of pelvic floor muscle dysfunctions that describe the parameters used in the therapy of these women will also be included.

## Type of outcome measures

### Primary outcomes.

1. Baseline pelvic floor muscle tone

2. Maximum voluntary contraction of the pelvic floor muscles

3. Sustained contraction of the pelvic floor muscles

4. Functionality of the pelvic floor muscles (contraction-relaxation coordination capacity)

5. Functionality of the pelvic floor muscles (ability for rapid contractions)

### Secondary outcomes.

6. Types of electromyographic equipment

7. Patient positioning during the intervention

8. Limitations of the chosen therapy

9. Types of Female Pelvic Floor Dysfunction More

10. Adverse events

### Electronic database.

1. Cochrane Central Register of Controlled Trials (CENTRAL)

2. MEDLINE (Pubmed)

3. Banco de Dados de Evidências em Fisioterapia (PEDro)

4. Scopus

5. Web of Science

6. Scielo

7. US National Institutes of Health, Register of Continuous Trials, ClinicalTrials.gov (www. clinictrials.gov);

*Literature search strategies.* The strategy on the terms used in each database can be found in S1 Appendix.

**Searching other resources.**   Reference checking of primary studies will be done manually and review articles will be added to the references.

## Selection of studies

**Data collection and analysis.**   For each search strategy, two reviewers will independently (ACRL and S) evaluate the studies from the databases in the order: title, abstract and full reading. Eligible studies will be read in full and data extracted for inclusion. The exclusion reasons

**Table 1. Study characteristics related to the number of participants, inclusion, and exclusion criteria.**

| Author/ Year | Number of participants | Inclusion criteria | Exclusion criteria |
|---|---|---|---|

for the studies will also be analyzed one by one. Disagreements regarding articles will be resolved by a third author by casting a vote (ESRV).

Duplicate studies will be identified and excluded. Studies involving men, children or non-pregnant women will also be excluded from the eligibility process and detailed in the Guideline Prisma flowchart.

**Data extraction and management.** The authors will extract the characteristics below from the included studies based on the PICO acronym [22]. The PICO acronym stands for Patient, Intervention, Comparison, and Outcome; these four components are fundamental elements in evidence-based practice for formulating research questions and conducting literature searches (Table 1)

1. Participants (P): *Inclusion criteria*. Pregnant women in any gestational phase or postpartum women up to 12 months after delivery, postpartum time, mean age, gestational week, floor dysfunction for treatment, sample inclusion and exclusion criteria, and sample description. *Exclusion criteria*. Women with neurological disease, urogenital tract infections.

2. Intervention (I): type of electromyographic equipment, types of electrodes, type of comparator equipment (if any), use of combined therapy (if any), intervention time, electromyographic parameters used, description of alternative interventions (placebo, no intervention or other intervention).

3. Comparator (C): different techniques or absence of intervention

4. Results (O): primary and secondary studies that evaluated electromyographic parameters for treatment of pelvic floor muscle dysfunctions.

5. Notes: authors, year of publication, funding of studies, and notable conflicts of interest among authors, study design, session time, follow-up time, study location, patient positioning, PFM functionality assessment method, treatment method.

Two reviewers (ACRL and SORL) will perform the initial data extraction from the included studies after reading the full text and within the inclusion criteria. A "Summary of included studies" table will be created, informing the total number of studies. In case of disagreements in data extraction, a meeting will first be held for consensus on the extraction, and in the persistence of doubt, a third author will decide by casting a vote (ERSV). The review author (ACRL) will transfer the summarized data to the Systematic Reviews management program (RevMan 2014) in order to generate the study report and analysis of heterogeneity and the possibility of meta-analysis of the data.

In case of lack of important data to perform the analysis, the author (ACRL) will contact the authors to provide details of the study in question. A professional fluent in the English language or Google Translator will assist in the translation of other published languages in case of doubts. The main results will be carefully reanalyzed by the study authors after translation.

**Risk of bias assessment in the included studies.** Two independent authors (ACRL and SORL) will analyze the risk of bias of experimental studies using the Rob 2.0 tool and the Robins-I tool for non-experimental studies. Disagreements will be resolved by consensus or involving a third review author (ESRV).

The assessment of bias risk in experimental and quasi-experimental studies will be conducted according to the Cochrane Handbook of Intervention Systematic Reviews [23]. The selection of these study designs was based on the rationale that they provide therapeutic intervention data for a specific sample. For this research, the Rob 2.0 tools will be use for experimental studies, and Robins-I for non-experimental studies.

The Rob 2.0 tool is structured in five domains that have "signaling questions" with the possibility of answers in: "yes", "probably yes", "probably not", "no", "no information'" and "not applicable". Definitive "yes" and "no" answers often indicate that robust evidence is available. The "not applicable" option is only available for questions with a non-mandatory answer. The final score of the responses determines the risk of bias for each domain: "high risk of bias", "low risk of bias" or "unclear" [24].

The ROBINS-I tool evaluates seven domains of bias, classified by: low risk of bias, moderate risk of bias, severe risk of bias, critical risk of bias or no information. The result for the final analysis of each component of the domain is based on the answers to the guiding questions and tables that support the judgment of bias in each domain [25].

Other bias: the ROBINS-I tool also allows for ranking the overall risk of bias, which receives the least favorable ranking among the assessed risks for the assessment tool's domains.

**Evidence quality assessment.** The evidence quality will be analyzed using the GRADE tool [26]. The structure of the systematic review will follow the recommendation of the Preferred Reporting Items of Systematic Reviews and Meta-Analyses (PRISMA) guidelines [21] and the protocol is registered in the PROSPERO database [27], the international prospective register of systematic reviews in health and social care (www.crd.york.ac.uk/prospero).

**Assessment of bias during the systematic review.** The review must be conducted according to this protocol and any adjustments can be justified in the "Difference between protocol and review" tab in the systematic review session.

**Effect treatment measures.** Dichotomous data: the odds ratios (ORs) and their associated 95% confidence intervals (CIs) will be used to determine the value of dichotomous data.

Continuous data: will be evaluated using standardized mean differences (STDs) and their corresponding 95% CIs using the Mantel-Haenzel method.

In case of a difference between means, the standard mean can be used for studies that analyzed the same result using different methods. P-values less than 0.05 will be considered statistically significant in all cases.

**Issues related to a single analysis.** Data analyzed in the study may be analyzed using a single analysis on the outcomes found. Meta-analysis may be used for randomized studies only if the data are justified for doing so.

**Lost data.** The authors of the present study will be able to contact the authors of the articles listed for data analysis in order to resolve doubts or request numerical data that were not made explicit in the body of the text or which were not found in the Register of Continuous Tests, ClinicalTrials.gov (www.clinictrials.gov). There are cases where only the abstract of the study contains the information, however, in the course of the text it does not. The failure of contacts and loss of entered data cause serious bias and this will be considered in the GRADE system.

## Evaluation of heterogeneity

If great heterogeneity is identified between the studies, the possible causes of the data discrepancy can be evaluated using specific sub-groups for analysis. If it is not possible to analyze the subgroup, the qualitative analysis will be summarized and presented in tables. In this case, the $I^2$ statistic and the p-value obtained from the Cochrane's Q test can be used.

**Table 2. Study characteristics.**

| Authors, Year | Pregnancy participants | Postpartum participants | Gestational age (weeks) | EMG protocols | Outcomes | Conclusions |
|---|---|---|---|---|---|---|
| Statistical analysis for analysis of subgroup interactions will be analyzed using the Rev-Man tool. | | | | | | |

## Evaluation of the risk of bias

The funnel chart can investigate reported biases. Symmetry of the funnel plot can be visually evaluated and explored. The analysis will be performed using the Egger's test.

## Synthesis of the data

A random effects-model and sensitivity analysis with a fixed model can be used for subgroup analysis and data heterogeneity. It is suggested to follow the results below for subgroup analysis:

1. Electromyographic parameters for the treatment of stress urinary incontinence in pregnancy and postpartum women.

2. Limitations of the chosen therapy

   The following results will be used in subgroup analyzes:

1. Functionality of the pelvic floor muscles

2. Most common types of female pelvic floor dysfunction

3. Adverse events

   The data synthesis of the analyzed studies will be allocated into the following sections: study characteristics (design, method of randomization (if applicable), blinding, allocation concealment, statistical methods); participants (pregnant and/or postpartum women); interventions (use of sEMG); clinical outcomes (types of outcomes measured: dichotomous or continuous, and adverse effects) (Table 2).

## Sensitivity analysis

A sensitivity analysis will be conducted to obtain a solid conclusion and to evaluate the stability of the results. All analyses will be performed using STATA SE 14.0. The sensitivity analysis may explore the influence of the quality of results. This can be assessed by excluding studies at high risk of bias.

## Discussion

The literature suggests that sEMG, with or without TMAP, can be employed as an effective treatment for pelvic floor dysfunctions, such as SUI. Although clinical trials in the literature provide electromyographic parameters for the treatment of PFMD in pregnant and postpartum women, these data are not summarized. This review aims to analyze the various applied protocols, methods, and terminologies used.

To date, this will be the first systematic review to examine electromyographic patterns in pregnant and postpartum women. These results will investigate whether there is scientific support for clinical practice in the use of electromyography in this population.

## Supporting information

**S1 Checklist. PRISMA-P (Preferred Reporting Items for Systematic review and Meta Analysis Protocols) 2015 checklist: Recommended items to address in a systematic review protocol.**
(DOCX)

**S1 Appendix.**
(DOCX)

## Author Contributions

**Conceptualization:** Alethéa Cury Rabelo Leitão, Silvia Oliveira Ribeiro Lira, Elizabel de Souza Ramalho Viana.

**Data curation:** Alethéa Cury Rabelo Leitão.

**Formal analysis:** Alethéa Cury Rabelo Leitão, Silvia Oliveira Ribeiro Lira, Elizabel de Souza Ramalho Viana.

**Investigation:** Alethéa Cury Rabelo Leitão.

**Methodology:** Alethéa Cury Rabelo Leitão, Silvia Oliveira Ribeiro Lira, Elizabel de Souza Ramalho Viana.

**Project administration:** Alethéa Cury Rabelo Leitão.

**Resources:** Alethéa Cury Rabelo Leitão, Elizabel de Souza Ramalho Viana.

**Software:** Alethéa Cury Rabelo Leitão.

**Supervision:** Alethéa Cury Rabelo Leitão, Silvia Oliveira Ribeiro Lira, Elizabel de Souza Ramalho Viana.

**Validation:** Alethéa Cury Rabelo Leitão, Elizabel de Souza Ramalho Viana.

**Visualization:** Alethéa Cury Rabelo Leitão, Silvia Oliveira Ribeiro Lira, Elizabel de Souza Ramalho Viana.

**Writing – original draft:** Alethéa Cury Rabelo Leitão, Silvia Oliveira Ribeiro Lira, Elizabel de Souza Ramalho Viana.

**Writing – review & editing:** Alethéa Cury Rabelo Leitão, Elizabel de Souza Ramalho Viana.

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
