## [Decision Letter · Decision Letter 0]

26 Apr 2024

PONE-D-23-43712Electromyographic parameters for treating pelvic floor disorders in pregnant and postpartum women: A review protocolPLOS ONE

Dear Dr. LEITAO,

Thank you for submitting your manuscript to PLOS ONE. After careful consideration, we feel that it has merit but does not fully meet PLOS ONE’s publication criteria as it currently stands. Therefore, we invite you to submit a revised version of the manuscript that addresses the points raised during the review process.

We look forward to receiving your revised manuscript.

Kind regards,

Mohammed Usman Ali

Academic Editor

PLOS ONE

Journal Requirements:

2. Thank you for stating the following financial disclosure: "NO

The funders did not and will not have a role in study design, data collection and analysis, decision to publish, or preparation of the manuscript."

Reviewers' comments:

Reviewer's Responses to Questions

**Comments to the Author**

1. Does the manuscript provide a valid rationale for the proposed study, with clearly identified and justified research questions?

Reviewer #1: No

Reviewer #2: Partly

Reviewer #3: Partly

2. Is the protocol technically sound and planned in a manner that will lead to a meaningful outcome and allow testing the stated hypotheses?

Reviewer #1: No

Reviewer #2: Yes

Reviewer #3: Partly

3. Is the methodology feasible and described in sufficient detail to allow the work to be replicable?

Reviewer #1: No

Reviewer #2: Yes

Reviewer #3: Yes

4. Have the authors described where all data underlying the findings will be made available when the study is complete?

Reviewer #1: No

Reviewer #2: Yes

Reviewer #3: Yes

5. Is the manuscript presented in an intelligible fashion and written in standard English?

Reviewer #1: No

Reviewer #2: Yes

Reviewer #3: No

6. Review Comments to the Author

You may also provide optional suggestions and comments to authors that they might find helpful in planning their study.

Reviewer #1: Major Revisions Needed:

This systematic review protocol requires major revisions to meet reporting standards before it can be considered acceptable for publication. Key details are lacking on eligibility criteria, search methods, data collection/analysis, and risk of bias assessment.

Specific comments:

The eligibility criteria require significant expansion to specify in detail the study designs, patient populations, interventions, comparators, and outcome measures to be included.

The search methods need to be described in greater depth - search terms, databases, dates searched, and a full planned search strategy in each database should be provided.

Details on data extraction, risk of bias assessment, evidence synthesis, meta-analysis, and evaluation of certainty of evidence using GRADE are currently inadequate and need considerable bolstering to meet systematic review standards.

The protocol follows future tense, but should be revised to past tense as it reports planned methods.

The introduction requires expansion with more background on pelvic floor disorders in the populations of interest and rationale for the review. The objectives and knowledge gap being addressed should also be more clearly specified.

The overall reporting needs to thoroughly follow PRISMA-P and Cochrane systematic review protocol guidelines. Registration in PROSPERO is good, but substantial enhancements of reported methods are required.

I recommend thoroughly revising the manuscript to address these major gaps in key systematic review methods and better adhere to reporting standards before resubmitting. Please feel free to contact me for any clarification or guidance needed on strengthening your protocol. Properly detailing your systematic review plans is an essential precursor to conducting a rigorous review.

Reviewer #2: This study surveys Electromyographic parameters for treating pelvic floor disorders in pregnant and postpartum women: A review protocol.

It is an interesting and new topic. The comments below may help you rewrite the article.

- In the title: “Electromyographic parameters for treating pelvic floor disorders in pregnant........”, do you want to survey EMG parameters or treatment effects?

- Please determine whether you want to include studies that used electromyography as an intervention or assessment. These two types of studies will be different from each other. You wrote intervention somewhere in the manuscript and evaluation somewhere else. Please clarify this issue. It would be better to exclude studies that only evaluated electromyography and include studies that used EMG biofeedback as a treatment.

- Which type of pelvic floor disorders would you like to include? Urinary (incontinence, retention...), bowel (constipation, incontinence…), or sexual disorders? These are different from each other and have different treatment protocols. I think it is better to limit the inclusion criteria.

- What do MAPMD in line 131 and APMD in line 269 stand for?

Reviewer #3: Abstract

You abstract is clear and gives a reader an understanding of what to expect in the article.

Introduction

Description of the condition

Please follow usual approaches to writing an introduction. Consider reading published articles from PlosONE to see how their introductions are written.

This is a good attempt but its largely descriptive; which may be fine. However, you need to give your reader an understanding of the burden (whether globally or nationally) to warrant further review or exploration of the evidence. Also, your subheading reads 'description of condition', however, you are also describing PFMT. Is this standard practice? Consider avoiding subheading or if you decide to use them, stay within the remit of the subheading. Either way, please make your introduction more coherent to build a case for the review.

Description of intervention

The rationale for this section is not clear. Are you trying to identify a gap within the evidence to make a case for your review? If so you need to be explicit about the gaps and the start to make a case for further studies. Again, this section did not describe an intervention but only reported previous reviews that have applied certain interventions. Those interventions need to be described based on your subheading.

line 83 - It will be prudent to report the number of studies that were included in the meta-analysis rather than the studies that were retrieved on initial search.

line 86 - What were the effect sixes and p-values?

How the intervention will work

Are you making the argument that studies do not report optimal parameters for electromyography? If so, how do you determine that there should be an optimal dosage? Also, what is the significance of EMG since it is not the treatment but a means of monitoring? Will the monitoring with EMG change the parameters regarding conservative treatment? If not what will be the value of knowing the optimal parameters for the EMG.

Is it the case that there are optimal parameters for conservative management for pelvic floor disorders and the can be used to monitor the muscle activity and improvement?

You just need to be clear regarding what the value of knowing the parameters of EMG for conservative management of pelvic floor disorders.

Why is this review important?

My understanding is that EMG monitors muscular activity to guide diagnosis of muscular disorders. In that sense, it can also guide or demonstrates muscles that are improving with management. I wonder what an optimal parameter will mean in terms of a muscular disorder? What grade of injury to the muscle will you consider? You need a strong justification for your review.

Materials and Methods

line 127 - Will EMG be used in treatment or monitor improvement as a result of treatment?

line 129 - Check your spellings

line 131 - What is MAPMD? Have you introduced and explained this?

line 171 - Please check you tenses

line 176 - How did you determine this data extraction procedure? Is it based on evidence?

Are there evidence-based data extraction tools you can apply?

line 182 - Is EMG an intervention or a diagnostic tool?

line 202 - Who is you?

line 208 - You need to justify the basis for considering experimental and non-experimental studies.

Which of your study designs target non-experimental studies? This is unclear. How do you plan to use different tools and synthesize the information.

line 260 - How can you perform a qualitative analysis of quantitative data? This is unclear.

line 263 - How will this process be appropriate for non-experimental studies?

line 274 - How will this process be applied to non-experimental studies?

Discussion

I am not convinced about the need for this review.

If you are using EMG as a diagnostic tool and also guide conservative management, you need to make it clear what the value of knowing the parameters on the EMG per patient. Will this be used as a target parameter for management? Will that be possible to determine given that various factors may contribute to recovery, peoples' body constitution may impact readings, and level of muscle disorder can contribute to parameter determination. Please consider these and revise the protocol accordingly. This is a very good attempt.

7. PLOS authors have the option to publish the peer review history of their article (what does this mean?). If published, this will include your full peer review and any attached files.

Reviewer #1: No

Reviewer #2: No

Reviewer #3: No

---

## [Author Response · Author response to Decision Letter 0]

3 Jun 2024

Dear Reviewers,

Ref.: Manuscript "Electromyographic parameters for treatment pelvic floor disorders in pregnant and postpartum women: A review protocol"

We appreciate the valuable suggestions for correcting the manuscript. We believe that these revisions have significantly improved our manuscript, making it clearer and more comprehensive. We are confident that the changes made address the reviewers' comments and enhance the quality and robustness of our research.

Thank you again for your consideration and we look forward to your positive response. We are available for any further questions or clarifications

RESPONSE TO REVIEWERS

REVIEW 1

ABSTRACT

Question 1: The abstract needs more detail on the methods - what type of studies will be included, how will they assess risk of bias, etc. Currently it only mentions the databases that will be searched. 

Response 

The section was rewritten detailed the types of studies to be included, the assessment of risk of bias using the Cochrane tool and the mention of the study's databases.

Question 2: The objective should be more clearly stated - will this review analyze the efficacy of EMG parameters for treating pelvic floor disorders in these populations, or just describe the parameters used? This is unclear. 

Response 

The text was rewritten to emphasize the main objective of the research: the treatment of pelvic floor dysfunctions using sEMG (p. 6). Sections mentioning pelvic floor assessment were removed. In the “Type of studies” and “Type of interventions” sections RCTs and non-RCTs using sEMG as a treatment were specified (p. 7). 

INTRODUCTION

Question 1: The introduction could benefit from more detail on pelvic floor disorders in pregnancy/postpartum and how EMG may help in diagnosis and treatment. Currently it is very brief. 

Response

The introduction has been rewritten following models from previous publications in the journal PLOS ONE (Barbosa et al., 2020[1]; Lira et al., 2022[2] ; Zielinski et al., 2023[3]). Additional information has been included on the epidemiology of dysfunctions, the effects of pathologies on women's health, the different presentations of dysfunctions and electromyography treatment in pregnant women who have had vaginal deliveries. In the end, the importance of the study has been more thoroughly detailed(p.3-7).

Question 2: More rationale is needed on why this review is important - what gap in knowledge is it addressing? How could it improve clinical practice? 

Response

The section was rewritten to address the issue comprehensively. The role of sEMG in treating common dysfunctions within the study population such as stress urinary incontinence and pudendal nerve injury during vaginal delivery was highlighted. The importance of the current review as a contribution to defining parameters used in the treatment of pelvic floor dysfunctions in pregnant and postpartum women was emphasized(p.5-6).

METHODS

Question 1: The eligibility criteria need significant expansion - what study designs will be included and excluded? What patient population characteristics, types of interventions, comparators, and outcome measures? 

Response

The steps for data extraction and management were detailed based on the PICO acronym. Each letter was selected to better formulate the research question and guide the reader through the steps for the search strategy of eligible articles necessary for data extraction. The following table was added at the end (p.8).

PICO:P (Population): Specifies the population or patient group of interest, I (Intervention): Defines the intervention being studied, C (Comparison): Identifies the comparison group or treatment, O (Outcome): Describes the outcomes of interest.

Question 2: The search strategy requires more detail - search terms, dates searched, and full planned search in each database should be provided. 

Response

The search strategy was included in S1 Appendix, highlighting the descriptors to be used and the Boolean operators with their combinations. (electromyography OR "electric myography " OR "electrical myography " OR "quantitative electromyography " OR "electromyographic examination") AND ("pelvic floor disorder" OR "pelvic diaphragm disfunction" OR "pelvic floor dysfunction" OR "pelvic floor disorders")

Question 3: Details on the data extraction process, risk of bias assessment, analysis/synthesis, etc. are currently inadequate and require considerable additions following systematic review reporting standards. 

Response

The assessment of bias risk in experimental and quasi-experimental studies will be conducted according to the Cochrane Handbook of Intervention Systematic Reviews. The selection of these study designs was based on the rationale that they provide therapeutic intervention data for a specific sample. For this research, the Rob 2.0 tools will be used for experimental studies, and Robins-I for non-experimental studies(p.9)

Question 4: There are no details on evaluation of certainty/quality of evidence. This is an essential component.

Response

The evidence quality will be analyzed using the GRADE tool. The structure of the systematic review will follow the recommendation of the Preferred Reporting Items of Systematic Reviews and Meta-Analyses (PRISMA) guidelines and the protocol is registered in the PROSPERO database , the international prospective register of systematic reviews in health and social care (www.crd.york.ac.uk/prospero) (p.10).

Question 5: Overall, the methods lack sufficient detail at present for a systematic review protocol. There are several key items missing that need to be expanded following PRISMA-P and Cochrane guidelines. 

Response

The section was rewritten to justify the selection of the study types, and consequently, the tools for assessing the risk of bias in the included studies were detailed. Regarding other risks of bias and the synthesis of information (p.12).

RESULTS

Question 1: This section is underdeveloped since the review has not yet been conducted. But some anticipated analysis/synthesis details could be added. 

Response

The sectrion will find in S1File. We anallysed PRISMA-P (Preferred Reporting Items for Systematic review and Meta-Analysis Protocols) 2015 check list: Recommended items to address in a systematic review protocol)

DISCUSSION 

Question 1: The current discussion is very speculative. This should be revised once findings are available to discuss limitations, implications, conclusions, etc. 

Response 

The discussion section was rewritten to emphasize the gap in the scientific literature on the topic, the clinical implications, and the importance of the review's results for professionals using surface electromyography in their clinical practice (p.12). 

REVIEW 2

Question 1: In the title: “Electromyographic parameters for treating pelvic floor disorders in pregnant........”, do you want to survey EMG parameters or treatment effects?

Response

The research aims to analyze the electromyographic parameters used in the treatment of pelvic floor dysfunctions.

Question 2: Please determine whether you want to include studies that used electromyography as an intervention or assessment. These two types of studies will be different from each other. You wrote intervention somewhere in the manuscript and evaluation somewhere else. Please clarify this issue. It would be better to exclude studies that only evaluated electromyography and include studies that used EMG biofeedback as a treatment.

Response

The text was rewritten to emphasize the main objective of the research: the treatment of pelvic floor dysfunctions using sEMG (p. 6). Sections mentioning pelvic floor assessment were removed. In the “Type of studies” and “Type of interventions” sections, RCTs and non-RCTs using sEMG as a treatment were specified (p. 7). 

Question 3: Which type of pelvic floor disorders would you like to include? Urinary (incontinence, retention...), bowel (constipation, incontinence…), or sexual disorders? Response 

These are different from each other and have different treatment protocols. I think it is better to limit the inclusion criteria.

Question 4: Where was it written “electromyography in pregnant or postpartum women for evaluation or treating pelvic floor dysfunctions (IUE) which analyzed baseline tone, contraction and relaxation capacity.”

Response

It has been corrected to “sEMG in pregnant or postpartum women for treating pelvic floor dysfunctions (IUE) which analyzed baseline tone, contraction and relaxation capacity.”

Question 5: What do MAPMD in line 131 and APMD in line 269 stand for?

Response

MAPMD is the acronym for "disfunção dos músculos do assoalho pélvico" in the native language (Portuguese). The term has been translated to English as "pelvic floor muscle dysfunction (PFMD)." Where was it written MAPMD, it has been corrected to PFMD.

REVIEW 3

DESCRIPTION OF THE CONDITION

Question 1: Line 57: Please follow usual approaches to writing an introduction. Consider reading published articles from PlosONE to see how their introductions are written

Question 2: Line 63: This is a good attempt but its largely descriptive; which may be fine. However, you need to give your reader an understanding of the burden (whether globally or nationally) to warrant further review or exploration of the evidence. Also, your subheading reads 'description of condition', however, you are also describing PFMT. Is this standard practice? Consider avoiding subheading or if you decide to use them, stay within the remit of the subheading. Either way, please make your introduction more coherent to build a case for the review.

Response to line 57 e 63

The introduction has been rewritten following models from previous publications in the journal PLOS ONE (Barbosa et al., 2020[1]; Lira et al., 2022[2] ; Zielinski et al., 2023[3]). Additional information has been included on the epidemiology of dysfunctions, the effects of pathologies on women's health, the different presentations of dysfunctions and electromyography treatment in pregnant women who have had vaginal deliveries. In the end, the importance of the study has been more thoroughly detailed(p.3-7).

.

DESCRIPTION OF THE INTERVENTION

Question 1: Line 76: The rationale for this section is not clear. Are you trying to identify a gap within the evidence to make a case for your review? If so you need to be explicit about the gaps and the start to make a case for further studies. Again, this section did not describe an intervention but only reported previous reviews that have applied certain interventions. Those interventions need to be described based on your subheading. 

Response:

The justification for the study has been rewritten with more detail. In the first two paragraphs, the description of the intervention begins by defining surface electromyography (sEMG), its mode of use and a description of the intervention. The gaps in the literature have been better described through studies on different findings. Finally, the data that are important to investigate in the treatment using EMG in pregnant and postpartum women have been presented.

Question 2: Line 83: It will be prudent to report the number of studies that were included in the meta-analysis rather than the studies that were retrieved on initial search.

Response

The meta-analysis by Nunes et al. (2019) was described in greater detail (11 studies). A new systematic review by Chmielewska et al. (2019), including a sample of 12 studies, was added to the text.

Question 3: Line 86: What were the effect sixes and p-values?

Response

The systematic review by Mateus-Vasconcelos et al. (2018) was detailed further regarding the characteristics analyzed by electromyography, which can be investigated in the scientific research. The meta-analysis by Nunes et al. (2019) was better described in the text, highlighting the use of BFB-EMG in the training of pelvic floor muscles in women with SUI. Consequently, the effects of the study objectives were detailed, and the current topic's objective was discussed. The studies show a p-value < 0.05

HOW THE INTERVENTION WILL WORK

Question 1: Line 88: Are you making the argument that studies do not report optimal parameters for electromyography? If so, how do you determine that there should be an optimal dosage? Also, what is the significance of EMG since it is not the treatment but a means of monitoring? Will the monitoring with EMG change the parameters regarding conservative treatment? If not what will be the value of knowing the optimal parameters for the EMG. Is it the case that there are optimal parameters for conservative management for pelvic floor disorders and the can be used to monitor the muscle activity and improvement? You just need to be clear regarding what the value of knowing the parameters of EMG for conservative management of pelvic floor disorders.

Response

The section was rewritten with more details about EMG and its characteristics. Initially, electromyography was defined and its utility in aiding healthcare professionals in clinical practice was explained. Additionally, the characteristics important for decision-making in the treatment of pelvic floor dysfunctions in pregnant and postpartum women were described. This study will contribute to the production of more scientific evidence on the subject.

WHY IS THIS REVIEW IMPORTANT?

Question 1: Line 116: My understanding is that EMG monitors muscular activity to guide diagnosis of muscular disorders. In that sense, it can also guide or demonstrates muscles that are improving with management. I wonder what an optimal parameter will mean in terms of a muscular disorder? What grade of injury to the muscle will you consider? You need a strong justification for your review.

Response

The section was rewritten to address the issue comprehensively. The role of sEMG in treating common dysfunctions within the study population such as stress urinary incontinence and pudendal nerve injury during vaginal delivery was highlighted. The importance of the current review as a contribution to defining parameters used in the treatment of pelvic floor dysfunctions in pregnant and postpartum women was emphasized.

TYPE OF STUDIES

Question 1: Line 127: Will EMG be used in treatment or monitor improvement as a result of treatment?

Response

Where was it written “electromyography in pregnant or postpartum women for evaluation or treating pelvic floor dysfunctions (IUE) which analyzed baseline tone, contraction and relaxation capacity.”, it has been corrected to “sEMG in pregnant or postpartum women for treating pelvic floor dysfunctions (IUE) which analyzed baseline tone, contraction and relaxation capacity.”

TYPE OF PARTICIPANTES

Question 1: Line 129: Check your spellings

Response

Where was it written “including.”, it has been corrected to “will be included.”

Question 2: Line 131: O que é a MAPMD? Já o apresentou e explicou?

Response

MAPMD is the acronym for "disfunção dos músculos do assoalho pélvico" in the native language (Portuguese). The term has been translated to English as "pelvic floor muscle dysfunction (PFMD)."Where was it written MAPMD, it has been corrected to PFMD.

SELECTION OF STUDIES

Question 1: Line 171: Please check you tenses

Response

Where was it written “Disagreements regarding articles were resolved by a third author by casting vote (ESRV)”, it has been corrected to disagreements regarding articles will be resolved by a third author by casting vote (ESRV)

DATA EXTRACTION AND MANAGEMENT

Question 1: Line 176: How did you determine this data extraction procedure? Is it based on evidence? Are there evidence-based data extraction tools you can apply? 

Response

The steps for data extraction and management were detailed based on the PICO acronym. Each letter was selected to better formulate the research question and guide the reader through the steps for the search strategy of eligible articles necessary for data extraction. The following table was added at the end (p.8).

PICO:P (Population): Specifies the population or patient group of interest, I (Intervention): Defines the intervention being studied, C (Comparison): Identifies the comparison group or treatment, O (Outcome): Describes the outcomes of inter

---

## [Decision Letter · Decision Letter 1]

13 Aug 2024

PONE-D-23-43712R1Electromyographic parameters for treatment of pelvic floor disorders in pregnant and postpartum women: A review protocolPLOS ONE

Dear Dr. LEITAO,

Thank you for submitting your manuscript to PLOS ONE. After careful consideration, we feel that it has merit but does not fully meet PLOS ONE’s publication criteria as it currently stands. Therefore, we invite you to submit a revised version of the manuscript that addresses the points raised during the review process.

We look forward to receiving your revised manuscript.

Kind regards,

Shabnam ShahAli, Ph.D.

Academic Editor

PLOS ONE

Journal Requirements:

Reviewers' comments:

Reviewer's Responses to Questions

**Comments to the Author**

1. Does the manuscript provide a valid rationale for the proposed study, with clearly identified and justified research questions?

Reviewer #2: Yes

Reviewer #4: Yes

2. Is the protocol technically sound and planned in a manner that will lead to a meaningful outcome and allow testing the stated hypotheses?

Reviewer #2: Yes

Reviewer #4: Yes

3. Is the methodology feasible and described in sufficient detail to allow the work to be replicable?

Reviewer #2: Yes

Reviewer #4: Yes

4. Have the authors described where all data underlying the findings will be made available when the study is complete?

Reviewer #2: Yes

Reviewer #4: Yes

5. Is the manuscript presented in an intelligible fashion and written in standard English?

Reviewer #2: Yes

Reviewer #4: No

6. Review Comments to the Author

You may also provide optional suggestions and comments to authors that they might find helpful in planning their study.

Reviewer #2: Most of the requested items have been modified and implemented

Question 3 has not been answered. It is better for the authors to pay attention to this issue or to use the sub-group analysis for each type of pelvic floor dysfunction.

Reviewer #4: thank you for the opportunity to review this manuscript. All comments have been addressed, however the English language revision is needed.

7. PLOS authors have the option to publish the peer review history of their article (what does this mean?). If published, this will include your full peer review and any attached files.

Reviewer #2: No

Reviewer #4: **Yes: **Ghazal Kharaji

---

## [Author Response · Author response to Decision Letter 1]

17 Aug 2024

RESPONSE TO REVIEWERS

Dear Reviewers:

 We hope this email finds you well.

We would like to express our gratitude once again for the opportunity to revise our work, and we hope that the new version of the manuscript meets the expectations of the journal.

Below, we highlight the changes made in response to considerations. We remain available for any further clarifications that may be required.

Best Regards,

Alethéa Cury Rabelo Leitão

Federal University of Rio Grande do Norte

Department of Physical Therapy

JOURNAL REQUIREMENTS:

Question: 

Please review your reference list to ensure that it is complete and correct. If you have cited papers that have been retracted, please include the rationale for doing so in the manuscript text, or remove these references and replace them with relevant current references. Any changes to the reference list should be mentioned in the rebuttal letter that accompanies your revised manuscript. If you need to cite a retracted article, indicate the article’s retracted status in the References list and and also include a citation and full reference for the retraction notice.

Response 

We have replaced references 2, 4, 8, 9, and 15. All references have been updated without compromising the scientific content.

Reference 2:

Before:

Caetano AS, Gomes C, Fernandes C, Baena H, Lopes DM. Incontinência urinária e a prática de atividades físicas. Rev Bras Med Esporte. 2007;13(4):270–4.

After:

DeLancey JOL, Masteling M, Pipitone F, LaCross J, Mastrovito S, Ashton-Miller JA. Pelvic floor injury during vaginal birth is life-altering and preventable: what can we do about it? Am J Obstet Gynecol. 2024 Mar;230(3):279-294.e2. doi: 10.1016/j.ajog.2023.11.1253. Epub 2024 Jan 2. PMID: 38168908; PMCID: PMC11177602.

Reference 4:

Before:

Mørkved S, Bø K. Effect of pelvic floor muscle training during pregnancy and after childbirth on prevention and treatment of urinary incontinence: A systematic review. Br J Sports Med. 2014;48(4):299–310.

After:

Woodley SJ, Lawrenson P, Boyle R, Cody JD, Mørkved S, Kernohan A, Hay-Smith EJC. Pelvic floor muscle training for preventing and treating urinary and faecal incontinence in antenatal and postnatal women. Cochrane Database Syst Rev. 2020 May 6;5(5):CD007471. doi: 10.1002/14651858.CD007471.pub4. PMID: 32378735; 

Reference 8:

Before:

Paula A, Resende M, Nakamura MU, Alves E, Ferreira G, Petricelli CD, et al. Eletromiografia de superfície para avaliação dos músculos do assoalho pélvico feminino: revisão de literatura. 2011;18(3):292–7.

After:

de Oliveira Ferro JK, Lemos A, de Santana Chagas AC, de Moraes AA, de Oliveira-Souza AIS, de Oliveira DA. Techniques for Registration of Myoelectric Activity of Women's Pelvic Floor Muscles: A Scoping Review. Int Urogynecol J. 2024 May;35(5):947-954. doi: 10.1007/s00192-024-05744-0. Epub 2024 Mar 12. PMID: 38472341.

Reference 9:

Before:

Batista RLA, Franco MM, Naldoni LM V., Duarte G, Oliveira AS, Ferreira CHJ. Biofeedback and the electromyographic activity of pelvic floor muscles in pregnant women. Brazilian J Phys Ther [Internet]. 2011 Oct [cited 2015 Jun 13];15(5):386–92. 

After:

Błudnicka M, Piernicka M, Kortas J, Bojar D, Duda-Biernacka B, Szumilewicz A. The influence of one-time biofeedback electromyography session on the firing order in the pelvic floor muscle contraction in pregnant woman-A randomized controlled trial. Front Hum Neurosci. 2022 Sep 29;16:944792. doi: 10.3389/fnhum.2022.944792. PMID: 36248694;.

Reference 15:

Before:

Marques J, Botelho S, Pereira LC, Lanza AH, Amorim CF, Palma P, Riccetto C. Pelvic floor muscle training program increases muscular contractility during first pregnancy and postpartum: electromyographic study. Neurourol Urodyn. 2013 Sep;32(7):998-1003. doi: 10.1002/nau.22346. Epub 2012 Nov 5. PMID: 23129397.

After:

Botelho S, Pereira LC, Marques J, Lanza AH, Amorim CF, Palma P, Riccetto C. Is there correlation between electromyography and digital palpation as means of measuring pelvic floor muscle contractility in nulliparous, pregnant, and postpartum women? Neurourol Urodyn. 2013 Jun;32(5):420-3. doi: 10.1002/nau.22321. Epub 2012 Sep 28. PMID: 23023961. 

.

REVIEW 2

Question:

Most of the requested items have been modified and implemented. Question 3 has not been answered. It is better for the authors to pay attention to this issue or to use sub-group analysis for each type of pelvic floor dysfunction.

Response:

Before: "Electromyographic parameters for the treatment of APMD"

After: "Electromyographic parameters for the treatment of stress urinary incontinence in pregnancy and postpartum women."

The term "APMD" was replaced with "stress urinary incontinence in pregnancy and postpartum women" to clarify the specific dysfunction being investigated in this protocol.

REVIEW 4

Question:

All comments have been addressed; however, an English language revision is needed.

Response:

The English language revision has been completed. The Declaration of Revision is attached.

Best regards.

---

## [Editor Report · Decision Letter 2]

20 Aug 2024

Electromyographic parameters for treatment of pelvic floor disorders in pregnant and postpartum women: A review protocol

PONE-D-23-43712R2

Dear Dr. LEITAO,

We’re pleased to inform you that your manuscript has been judged scientifically suitable for publication and will be formally accepted for publication once it meets all outstanding technical requirements.

Kind regards,

Shabnam ShahAli, Ph.D.

Academic Editor

PLOS ONE
---

## [Editor Report · Acceptance letter]

28 Aug 2024

PONE-D-23-43712R2 

PLOS ONE

Dear Dr. Leitão, 

I'm pleased to inform you that your manuscript has been deemed suitable for publication in PLOS ONE. Congratulations! Your manuscript is now being handed over to our production team.

Kind regards, 

on behalf of

Dr. Shabnam ShahAli 

Academic Editor

PLOS ONE